# YOLOv4 with Deformable-Embedding-Transformer Feature Extractor for Exact Object Detection in Aerial Imagery

**DOI:** 10.3390/s23052522

**Published:** 2023-02-24

**Authors:** Yiheng Wu, Jianjun Li

**Affiliations:** College of Computer and Information Engineering, Central South University of Forestry and Technology University, Changsha 410004, China

**Keywords:** aerial imagery, ultra-high spatial resolution orbital imagery, object detection, YOLOv4, vision transformer, deep learning

## Abstract

The deep learning method for natural-image object detection tasks has made tremendous progress in recent decades. However, due to multiscale targets, complex backgrounds, and high-scale small targets, methods from the field of natural images frequently fail to produce satisfactory results when applied to aerial images. To address these problems, we proposed the DET-YOLO enhancement based on YOLOv4. Initially, we employed a vision transformer to acquire highly effective global information extraction capabilities. In the transformer, we proposed deformable embedding instead of linear embedding and a full convolution feedforward network (FCFN) instead of a feedforward network in order to reduce the feature loss caused by cutting in the embedding process and improve the spatial feature extraction capability. Second, for improved multiscale feature fusion in the neck, we employed a depth direction separable deformable pyramid module (DSDP) rather than a feature pyramid network. Experiments on the DOTA, RSOD, and UCAS-AOD datasets demonstrated that our method’s average accuracy (mAP) values reached 0.728, 0.952, and 0.945, respectively, which were comparable to the existing state-of-the-art methods.

## 1. Introduction

Object detection in ultra-high spatial resolution aerial images aims to rapidly determine a target’s location and classify the target. Object detection in aerial images is widely used in numerous important fields, including the localization and identification of military targets [1], natural environment protection [2], disaster detection [3], and urban construction planning [4].

The application of convolutional neural networks to computer vision and object detection has made significant strides in recent years. Successively, superior object detection algorithms, including the R-CNN series [5,6,7,8], the YOLO series [9,10,11], and the SSD series [12,13,14], have been proposed. With the introduction of ViT [15], the transformer is widely utilized in the field of image processing. Recently, transformer-based detectors such as DETR [16] and DINO [17] have been proposed and have performed better in some detection tasks than CNN-based networks. With large annotated natural image datasets such as MS COCO [18] and PASCAL VOC [19], these CNN-based and transformer-based target detection algorithms have achieved astonishing results. However, the above models have poor detection performance with aerial images, which is due to the specificity of aerial images [20]. Firstly, aerial images are taken at altitudes ranging from several hundred meters to tens of kilometers due to differences in observation equipment, which results in diverse sizes of targets in the same category. Secondly, aerial images are usually taken by acquisition equipment from a top view. As a result, the background of aerial images is complex and contains a great deal of redundant information, causing severe interference in detection. Finally, since aerial images are captured from high altitudes, they contain an abundance of small targets. For instance, a car’s pixels might be 5 × 5 or even smaller. This makes it challenging to extract edge features from aerial images of targets and to distinguish targets from the background. These characteristics make aerial image object detection a unique and difficult problem.

In this research, we focus on addressing the issues mentioned above to enhance the target detection performance in aerial images. In addition, the speed of detection is a significant obstacle for detection algorithms, as target detection from satellite imagery is typically performed in real time. A one-stage object detection algorithm, exemplified by the YOLO series [9,10,11], combines object classification and localization into a single-stage regression problem, vastly enhancing the detection speed in comparison to a traditional two-stage object detection algorithm of localization followed by classification. Currently, the most advanced versions of YOLO are YOLOv4 [21], YOLOv5, and YOLOX [22]. Compared to YOLOv4, YOLOv5 and YOLOX utilize a large number of dense connection structures; their models are more complex, and their detection speed is decreased, but their detection accuracy is comparable. Therefore, we enhanced the feature extraction network and the feature fusion network based on YOLOv4 to meet the detection requirements for targets in optical images with ultra-high spatial resolutions, particularly small targets.

DET-YOLO is the name of the enhanced YOLOv4 variant. Experimental results showed that our proposed DET-YOLO outperformed the general YOLOv4 in detecting targets in aerial images.

This study’s contribution is summarized as follows:A new feature extraction network named a deformable embedding vision transformer (DEViT) is proposed, which has the excellent global feature extraction capability of transformers while using deformable embedding instead of normal embedding to efficiently extract features at different scales. In addition, a fully convolutional feedforward network (FCFN) is used to replace the feedforward neural network (FFN), and the extraction of location information is effectively enhanced by introducing zero padding and convolutional operations.The depthwise separable deformable convolution (DSDC) is proposed to reduce computational effort while preserving the deformable convolution’s ability to zero in on regions of interest. It is proposed that the depthwise separable deformable pyramid (DSDP) module extracts multiscale feature maps and prioritizes key features.Our proposed DET-YOLO achieved the highest accuracy among existing models, with mean average precision (mAP) values of 0.728 on the DOTA dataset, 0.952 on the RSOD dataset, and 0.945 on the UCAS-AOD dataset.

The rest of the paper is organized as follows: Section 2 describes the proposed network and the individual components; Section 3 presents the dataset and the evaluation metrics used; Section 4 gives the experimental results and analyses to demonstrate the accuracy and validity of the proposed algorithm; and Section 5 gives the conclusions.

## 2. Methodology

### 2.1. Review of YOLOv4

YOLOv4’s architecture comprises three primary components: the backbone, the neck, and the predicting head. The backbone extracts feature data from an image input. The neck collects and combines multiscale feature data to generate three distinct-scale feature maps. Based on these created feature maps, the predicting head detects objects. YOLOv4 employs the CSPDarknet53 [23] framework as its backbone, SPP [24] and PANet [25] as the neck, and YOLO’s detection head. In addition, YOLOv4 uses a number of measures to enhance the performance of target detection, including mosaic data enhancement, Mish activation functions, and CIOU loss.

### 2.2. DET-YOLO

Figure 1 depicts the architecture of our proposed DET-YOLO for object detection in aerial images. We employ DEViT as the backbone, DSDP and PANet as the neck, and YOLO’s detection head as the predicting head. In addition, it employs the Mish [26] activation function in place of the leaky ReLU [27] activation function, CIoU [28] loss in place of the ordinary IoU [29] loss, and the cosine annealing strategy [30] for learning rate attenuation to achieve more accurate detection results. In the subsequent subsections, we elaborate on DEViT and DSDP principles.

### 2.3. Deformable Embedding Vision Transformer

Figure 2 depicts the structure of a non-hierarchical adaptive transformer, DEViT, that is proposed to have an excellent global feature capture capability while efficiently and flexibly extracting target information. We made the following enhancements: First, we proposed deformable embedding to capture targets of varying sizes in aerial images. Second, we proposed fully convolutional feedforward networks to improve the location information extraction.

These two aspects are elaborated upon below.

#### 2.3.1. Deformable Embedding

Traditional visual transformers, including hierarchical models such as PVT [31] and the Swin Transformer [32] as well as non-hierarchical models such as ViT [15] and DEiT [33], use a fixed-size patch embedding based on the implicit assumption that the fixed image split design is appropriate for all images. Given the input image X ∈ ℝC×H×W, where *C*, *H*, and *W* represent the channel dimension, height, and width of the feature, respectively, a vanilla patch embedding splits X into *N* patches of size *P* × *P* (N=HW/P2). The patches are then connected to form a sequence of flattened 2D patches (X p ∈ ℝN × (P 2·C)), each of which represents a rectangular region of input image X.

Nevertheless, the scale of the information contained in aerial images varies greatly due to the different observation carriers and acquisition methods, and the typical embedding methods frequently do not fit well or even destroy the target feature information, as depicted in Figure 3. Therefore, a data-dependent sparse embedding strategy is required for the flexible construction of relevant features.

DCN [34] proposes a deformable convolution, which allows the convolution kernel to be adaptively focused on the target region by obtaining an offset, as opposed to being limited to a fixed size and shape of the perceptual field. In response, we propose deformable embedding, which obtains the ability to extract objects of different scales by adaptively acquiring an offset, as depicted in Figure 4b.

As depicted in Figure 4a, the above embedding process can also be viewed as downsampling X using a linear embedding layer with kernel size *P* and strike size *P* to obtain the feature map X′∈ ℝ (P 2·C) × HP × WP  and then reshaping the feature map X′ into X p. The process of embedding can be described as follows:(1)X′=Embed(X),
(2)Xp=Reshape(X′),
where *Embed*(∙) is the operation for linear embedding and *Reshape*(∙) is the flattening function.

For each patch *p*(*i*) on feature map X′, a rectangular region of the input image with the area P×P is represented. The coordinates of its lower left corner are denoted as $l_{i}^{1}=\left(x_{i}^{1}{,\;y}_{i}^{1}\right)$, and the sequence of coordinates corresponding to *p*(*i*) is denoted as $\left\{l_{i}^{1},\cdots ,l_{i}^{P\times P}\right\}$. These locations’ features are represented as $\left\{f_{\;i}^{\;j},\cdots ,f_{i}^{\;P\times P}\right\}$. Equation (3) illustrates how linear embedding flattens these features and processes them with a linear layer to produce a representation of the patch.
(3)$$p\left(i\right)=W_{patch}\cdot concat\left(f_{\;i}^{\;j},\cdots ,f_{i}^{\;P\times P}\right)+b_{patch}.$$

Feature map X is fed to the lightweight subnetwork *θ_offset_*(∙), which has a single convolutional layer with a 1 × 1 convolution kernel. The bias quantity feature map X*_offset_* ∈ ℝ2C×H×W is produced, and the layer 2*i*−1 and layer 2*i* channels represent the x- and *y*-axis offsets, respectively, for to the layer *i* channel of the original map. The offset associated with each position lij is ∆lij = *θ_offset_* (lij). Thus, Equation (4) depicts the representation of the new patch obtained via deformable embedding.
(4)p(i)=Wpatch·concat(f i j(li1+∆li1),⋯,fi P×P(liP×P+∆liP×P))+bpatch,
where *p*(*i*) constitutes the new feature graph X′. X′ is substituted into Equation (2) to obtain the deformable embedding result Xp. The procedure for *p* (*i*) is as follows:(5)Xp=concat(p(1),⋯,p(N)).

Our proposed deformable embedding receives a 2D offset on each pixel for each patch, allowing it to extract the target information adaptively and effectively avoiding the issue of semantic information corruption during vanilla splitting.

#### 2.3.2. Full Convolution Feedforward Network

A vanilla feedforward neural network (FFN) of visual transformers consists of two layers of MLP composition and a GELU [35] activation function, as demonstrated by Equation (6).
(6)FFN(F)=MLP(GELU(MLP(F))).

Due to the high computational cost of MLPs, they require a substantial amount of computation while effectively preserving the dimensionality of the feature channels. As shown in Figure 5, we propose an FCFN with enhancements to both reduce the computational effort and enhance the capability of feature extraction.

First, instead of MLP, we use two 1 × 1 depthwise separable convolutions [36] to improve the model’s robustness while further reducing the computational effort. Second, we enhance the network’s capacity to extract location data. The study by Islam et al. [37] demonstrated that the use of convolutional layers with zero padding can aid a network in learning absolute position data. Before the GELU activation function, we introduce a depth separable convolution with a stride of 1, a kernel size of 3, and a padding size of 1, which only marginally increases the computational effort. As shown in Equation (7), the combination of these two enhancements forms our fully convolutional neural network.
(7)$$\mathrm{FCFN}(F)=({\mathrm{DWConv1}}_{1\times 1}(\mathrm{GELU}({\mathrm{DWConv}}_{3\times 3}({\mathrm{DWConv1}}_{1\times 1}(F))))).$$

To visually demonstrate the improvement in the computational requirements of our FCFN over the FFN, we compare their computational costs and parameter counts.

A standard MLP layer accepts a $1\times M_{\mathrm{mlp}}$ feature sequence (*I*) as an input and outputs a $1\times N_{\mathrm{mlp}}$ feature sequence (*O),* where $M_{\mathrm{mlp}}$ and $N_{\mathrm{mlp}}$ represent the lengths of the input and output feature sequences, respectively. MLP contains $N_{\mathrm{mlp}}$ neurons (*n*). The standard MLP output feature sequence is computed as follows:(8)$$O_{n}=\sum_{i,j}n_{i}\cdot I_{j}.$$

The standard MLP incurs the following computational cost:(9)$${\cos t}_{\mathrm{mlp}}={\;M}_{\mathrm{mlp}}\cdot N_{\mathrm{mlp}}.$$

The standard MLP parameters are as follows:(10)$${\mathrm{param}}_{\mathrm{mlp}}={\;M}_{\mathrm{mlp}}\cdot {(N}_{\mathrm{mlp}}+1).$$

For FFN, the first MLP receives a $1\times {(P}^{\;2}\cdot C)$ feature sequence as an input and outputs a $1\times {(P}^{\;2}\cdot 4C)\;$feature sequence, while the second MLP receives a $1\times {(P}^{\;2}\cdot 4C)$ feature sequence as an input and outputs a ${(P}^{\;2}\cdot C)$ feature sequence. Therefore, the computational and parametric quantities of the FFN are shown in Equations (11) and (12), respectively.
(11)$$\begin{array}{c} {\mathrm{Cos}t}_{\mathrm{ffn}}={\;(P}^{\;2}\cdot C)\cdot {(P}^{\;2}\cdot 4C)+{(P}^{\;2}\cdot 4C)\cdot {(P}^{\;2}\cdot C)\\ =8\cdot P^{\;4}\cdot C^{\;2}, \end{array}$$
(12)$$\begin{array}{c} {\mathrm{param}}_{\mathrm{ffn}}={(P}^{\;2}\cdot C)\cdot {(P}^{\;2}\cdot 4C+1)+{(P}^{\;2}\cdot 4C)\cdot {(P}^{\;2}\cdot C+1)\\ =8\cdot P^{\;4}\cdot C^{\;2}+5\cdot P^{\;2}\cdot C. \end{array}$$

A standard convolutional layer receives an $M_{\mathrm{conv}}\times W\times H$ feature map (*D*) as an input and generates an $N_{\mathrm{conv}}\times A\times B$ feature map (*G*) as an output, where *W* and *H* represent the width and height of the input feature map, $M_{\mathrm{conv}}$ represents the number of input channels, *A* and *B* represent the width and height of the output feature map, and $N_{\mathrm{conv}}\;$represents the number of output channels. The convolution layer contains $N_{\mathrm{conv}}\;$convolution kernels ($K^{n}$) with an $M_{\mathrm{conv}}\times w\times h$ shape and a strike size of 1. The output feature sequence of a convolutional layer is computed as follows:(13)$$G_{k,l,n}=\sum_{i,j,m}K_{i,j,m}^{n}\cdot D_{k+i-1,l+j-1,m}.$$

A standard convolutional layer incurs the following computational expenses:(14)$${\cos t}_{\mathrm{conv}}=\;w\cdot h\cdot M_{conv}\;\cdot N_{\mathrm{conv}}\;\cdot \;A\cdot \;B.$$

The parameters of a standard convolutional layer are:(15)$${\mathrm{param}}_{\mathrm{conv}}=(w\cdot h\cdot N_{conv}+1)\cdot M_{conv}.$$

The computational cost of the depthwise separable convolution corresponding to the standard convolution is:(16)$${\cos t}_{\mathrm{dwconv}}=\;w\cdot h\cdot M_{conv}\;\cdot A\cdot {\;B+M}_{conv}\cdot N_{\mathrm{conv}}\;\cdot \;A\cdot \;B.$$

The computational cost of the depthwise separable convolution corresponding to the standard convolution is:(17)$${\mathrm{param}}_{\mathrm{dwconv}}=\;(w\cdot h+1)\cdot M_{conv}+{(M}_{conv}+1)\cdot N_{\mathrm{conv}}.$$

For the FCFN, the first DWConv takes a C×P×P feature map as an input and returns a 4C×P×P feature map; the second DWConv takes a 4C×P×P feature map as an input and returns a 4C×P×P feature map; and the third DWConv takes a 4C×P×P feature map as an input and returns a C×P×P feature map. Therefore, Equations (18) and (19) display the computational and parametric quantities of the FCFN.
(18)$$\begin{array}{c} {\cos t}_{\mathrm{fcfn}}=\;(1\cdot C\cdot P^{\;2}+4C\cdot C\cdot P^{\;2})+(9\cdot 4C\cdot P^{\;2}+4C\cdot 4C\cdot P^{\;2})+(1\cdot C\cdot P^{\;2}+4C\cdot C\cdot P^{\;2})\\ =24\cdot P^{\;2}\cdot C^{\;2}+41\cdot P^{\;2}\cdot C, \end{array}$$
(19)$$\begin{array}{c} {\mathrm{param}}_{\mathrm{fcfn}}=(1\cdot C\cdot P^{\;2}+C+4C\cdot C)+(9\cdot C+4C\cdot 4C\cdot P^{\;2})+(1\cdot C\cdot P^{\;2}+4C\cdot C\cdot P^{\;2})\\ =24\cdot C^{\;2}+17\cdot C. \end{array}$$

By substituting the FFN with the FCFN, the following reductions in computation and parameters are obtained:(20)$$\frac{{\cos t}_{\mathrm{fcfn}}}{{\cos t}_{\mathrm{ffn}}}=\frac{41}{{8P}^{\;2}C}+\frac{3}{P^{\;2}},$$
(21)$$\frac{{\mathrm{param}}_{\mathrm{fcfn}}}{{\mathrm{param}}_{\mathrm{ffn}}}=\frac{17+24C}{8P^{\;4}C+5P^{\;2}}.$$

DEViT employs a deformable embedding with a patch size of 16 and accepts high- resolution RGB satellite images as inputs; consequently, each FCFN is $\frac{113}{6144}$ of the computation and $\frac{89}{1574144}$ of the parameters of the corresponding standard FFN. As demonstrated in Section 4, the use of the FCFN rather than the FFN improves precision while significantly reducing the computational costs.

### 2.4. Depthwise Separable Deformable Pyramid

Instead of FPN, we propose a depthwise separable deformable pyramid to obtain feature maps at various scales from a non-hierarchical network architecture such as DEViT. The following subsections are descriptions of the DSDC and the DSDP, which comprises the DSDC.

#### 2.4.1. Depthwise Separable Deformable Convolution

DSDC is proposed by combining depthwise separable convolution [36] and deformable convolution [34], which significantly reduces the computational effort while adaptively focusing on the object region, as is shown in Figure 6.

By adding an offset convolution layer, the vanilla deformable convolution layer learns offsets from the previous feature mapping. Using a convolution kernel of the same size as the corresponding convolution and an offset feature map with an output space the same size as the input feature map with twice as many channels as the original feature map, the offset of the convolution kernel relative to the x and y axes on each pixel of each channel is determined using the offset convolution.

Thus, the deformation can concentrate on objects in a dense, adaptive, and local manner. However, the additional convolution operation requires a substantial amount of computation, which slows model inference and makes it difficult to meet the real-time requirement for aerial image object detection. Deformable convolution yields a spatially two-dimensional offset that is channel-independent and two-dimensional. Therefore, we can separate the calculation of the offset from the calculation of the channel, which drastically reduces the amount of computation required without affecting the effect.

A standard convolution is decomposed into a depthwise convolution and a pointwise convolution by depthwise separable convolution. For each input channel, the depthwise convolution performs a separate convolution operation. The pointwise convolution then applies a 1 × 1 convolution to combine the outputs of the depthwise convolution. A depthwise separable convolution reduces the computational effort by decomposing a standard convolution into a channel-by-channel convolution kernel and a point-by-point convolution.

A standard deformable convolution is decomposed by DSDC into a deformable depthwise convolution and a pointwise convolution. A deformable depthwise convolution learns the offset independently for each channel, whereas a pointwise convolution combines the output of the deformable depthwise convolution for each channel. DSDC replaces the depthwise convolution portion of the depthwise separable convolution with a deformable convolution to gain the ability to adaptively focus on the region of interest. DSDC strikes a balance between improved computational efficiency and the deformable convolution’s ability to extract features.

#### 2.4.2. Depthwise Separable Deformable Pyramid Module

Because DEViT employs a non-hierarchical transformer structure, it is only possible to obtain a single-scale feature map. Due to the varying scales of the targets, a multiscale feature acquisition capability is required in aerial images. The most prevalent method for constructing a multiscale feature map is a feature pyramid network, which employs a top-down architecture with skip connections to combine features at various levels. For non-hierarchical transformers, Li et al. [38] attempted to use a simple feature pyramid instead of a complex FPN structure, which effectively optimized the structure while maintaining accuracy.

We propose a simple multiscale feature acquisition module, dubbed a depthwise separable deformable pyramid, which only performs simple upsampling, downsampling, and a 1 × 1 convolution on the last layer of DEViT to generate different-scale feature maps. To focus the network on the region of interest and improve its ability to extract features, we introduce DSDC for each scale of features, causing the network to generate adaptive offsets for the target at each scale. Figure 7 depicts a comparison between DSDP and FPN.

Subsequent experiments revealed that DSDP performed better than FPN on DET-YOLOv4, which will be discussed in detail in the ablation experiments subsection in Section 4.

## 3. Datasets and Experimental Settings

### 3.1. Data Description

To validate the efficacy of DET-YOLO, we conducted experiments on the widely used public datasets RSOD [39], UCAS-AOD [40], and DOTA [41] for multiclass object detection in orbital images with high spatial resolutions.

The RSOD dataset includes 6950 annotated samples and 976 high-resolution RGB satellite images from Google Earth and Tianditu. The dataset includes four distinct types of objects, including an oil tank, an aircraft, an overpass, and a playground.

The UCAS-AOD dataset is a high-resolution aerial image dataset designed to detect small and medium-sized targets. It consists of 1510 images and 14,596 annotated instances with a fixed image size of 1280 × 659. The dataset is divided into two categories, aircraft and vehicles, and contains counterexamples with no instances of objects.

The DOTA-v1.5 dataset was derived from Google Earth, the JL-1 satellite, and the GF-2 satellite, among other platforms and sensors. It contains 2806 RGB images and 188,282 annotated instances with dimensions ranging from 800 × 800 to 4000 × 4000. It includes targets for 16 categories such as planes, ships, storage tanks, baseball diamonds, tennis courts, basketball courts, ground track fields, harbors, bridges, large vehicles, small vehicles, helicopters, roundabouts, soccer fields, swimming pools, and container cranes. Compared to version 1.0, DOTA-v1.5 adds a large number of objects smaller than 10 pixels, making object detection tasks more difficult.

To validate the generality of our model, we also conducted experiments on UAVDT [41]. Unlike the orbital-image-based dataset described above, UAVDT is a large-scale challenging benchmark dataset based on UAV images. It contains approximately 80,000 frames with annotated information. When used for object detection, the UAVDT dataset has three types of objects (cars, trucks, and buses) and contains 23,829 training images and 16,580 test images at a size of 1024 × 540. The UAV images have a higher spatial resolution compared to the orbital images. As shown in Table 1, objects smaller than 10 pixels are referred to as very small objects, objects larger than 10 pixels but smaller than 50 pixels are referred to as small objects, objects with a size between 50 and 300 pixels are referred to as medium objects, and objects larger than 300 pixels are referred to as large objects, according to the classification criteria of [42]. In the orbital image datasets, the DOTA dataset has the highest proportion of small objects and is the most difficult of the three datasets, as shown in Table 1. UCAS-AOD focuses on small to medium-sized objects, while RSOD is moderately challenging. UAVDT focuses on small objects. For input into the network, we maintained the original sizes of the RSOD dataset, UCAS-AOD, and UAVDT. We separated them into training and validation sets based on criteria outlined in [41,43,44]. For the DOTA dataset, the oversized images were cropped. Approximately 20,000 1024 × 1024 images were generated in total. There were 1414 training images and 939 validation images in the original DOTA dataset. We obtained 14,729 training images and 5066 test images after cropping. The breakdown of the dataset is detailed in Table 2.

### 3.2. Evaluation Metrics

As evaluation metrics, we chose precision (P), recall (R), average precision (AP), and mean average precision (mAP), which are frequently employed in target detection.

P and R are defined as:(22)$$P=\frac{TP}{TP+FP},$$
(23)$$R=\frac{TP}{TP+FN}.$$
where *TP* is true positive, *FP* is false positive, *TN* is true negative, and *FN* is false negative.

AP is defined as the area of the P-R curve created using precision and recall as follows:(24)$$\mathrm{AP}=\int_{0}^{1}P\left(R\right)dR,$$
where P(·) represents the P-R curve function. Using AP, we can precisely determine the model’s capture capability for various objects.

The mAP is the average of the APs for all categories and accurately reflects the model’s overall predictive performance. The definition is as follows:(25)$$\mathrm{mAP}=\frac{1}{K}\overset{K}{\underset{i=1}{\sum }}{\mathrm{AP}}_{i},$$
where *K* represents the number of categories and ${\mathrm{AP}}_{i}$ represents the mean precision of the i-th category.

Using the aforementioned indicators, we could evaluate our strategy effectively.

### 3.3. Implementation Details

Our proposed DET-YOLO was implemented in Windows 10 using the 1.8.1 version of the PyTorch framework. The experiments were conducted on a desktop computer equipped with an Intel i9-7920X processor and an NVIDIA RTX 2080 ti graphics processing unit with 11 GB of memory.

We proposed that DEViT serve as the backbone of DET-YOLO in place of CSPDarkNet53. As a pre-trained model for DEViT, we partially inherited the ViT model trained on ImageNet using the MAE method. According to [45], we used the SGD optimizer with an initial learning rate of 0.01 and a momentum of 0.937 for training. According to [46], DET-YOLO was trained for 50 epochs on the RSOD and UCAS-AOD datasets and 150 epochs on the DOTA dataset. According to the characteristics of the different datasets [47,48], the images in the DOTA dataset were input to the network at a size of 1024 × 1024, whereas the images in the RSOD and UCAS-AOD datasets were resized to 608 × 608 to improve the feature extraction ability of small targets while maintaining a small network scale. We kept the original dimensions of the images in the UAVDT dataset as an input to the model.

## 4. Experimental Results and Discussion

### 4.1. Contrasting Experiments

To determine the efficacy and universality of DET-YOLO, we compared it to the most representative methods on the DOTA, RSOD, and UCAS-AOD datasets. The experimental outcomes for three distinct datasets are shown in Table 3, Table 4, Table 5 and Table 6. Table 3 demonstrates that the mAP on the DOTA dataset was 0.728, which was an improvement of 0.012 over the suboptimal model SPH-YOLOv5, demonstrating that our model has excellent detection performance for small and very small targets. Table 4 demonstrates that our model achieved an mAP of 0.952 on RSOD, which was an improvement of 0.066 over the suboptimal model YOLOv5 and substantiated the efficacy of medium and large target detection. Table 5 demonstrates that our model achieved an mAP of up to 0.945 on UCAS-AOD, which was an improvement of 0.070 over the suboptimal model YOLOv5, demonstrating that our model has excellent detection performance for small and medium-sized targets, such as vehicles and airplanes. Table 6 demonstrates that our method achieved an mAP of up to 0.424 on the UAVDT dataset, which was an improvement of 0.005 over the suboptimal model YOLOv5, demonstrating that our method is applicable to UAV images. In conclusion, the detection performance of our proposed DET-YOLO for aerial targets of various sizes, particularly small targets, was superior to that of the current principal methods, demonstrating the efficacy of this method.

We plotted P-R curves and confusion matrices of DET-YOLO on the DOTA dataset to demonstrate the DET-YOLO detection performance for each category of the aerial targets. Figure 8 and Figure 9 show the P-R curve and the confusion matrix, respectively.

The AP of each category is the area below the corresponding P-R curve. The greater the AP, the better the detection performance for this category, and the larger the corresponding area. Each row of the confusion matrix represents an actual category of the sample, while each column represents the predicted category. The entire matrix reflects the model’s ability to classify the objectives of each category. The horizontal background represents FP, the negative sample proportion of the prediction error, and the vertical background represents FN, the positive sample proportion of the prediction error. In the confusion matrix, the diagonal data indicate the proportions of correctly classifiable categories. From the P-R curve, it can be seen that the prediction accuracy for container cranes was far lower than that of the other categories. Moreover, the confusion matrix in Figure 9a reveals that the FN value corresponding to container cranes was the highest, indicating a severe leakage phenomenon. This was because the number of training samples with container cranes was significantly lower than the other training samples, and there were only 774 in 970,170 instances, which made it difficult for the model to fit the corresponding features and led to the model’s poor feature extraction ability, making it difficult to effectively identify such targets. On the other hand, although the corresponding accuracy for small vehicles was great, the corresponding FP value was the highest, and there was a serious misdetection phenomenon. Small vehicles are small or extremely small targets, and there is a phenomenon known as stacking. Moreover, the cutting of the images caused some small vehicles in the dataset not to be marked, which further affected the FP value of the small vehicles. The confusion matrix in Figure 9b demonstrates that as the target became smaller, the prediction accuracy of the model decreased, with increasing leakage and misdetection.

We tested the reasoning speeds of different methods on the DOTA dataset, as shown in Table 7. Our method’s speed was between YOLOv4 and YOLOv5, as can be seen. DET-YOLO’s inference speed was slower than that of YOLOv4 because self-attentive operations are more computationally complex than convolutional operations. In addition, deformable embedding is a large kernel convolutional operation, which further slowed down DET-YOLO’s inference speed. Overall, our proposed DEViT required more computational resources and had greater feature extraction capability than YOLOv4’s CSPDarknet53. The more complex feature extraction network is the primary reason why DET-YOLO’s inference speed was slower than that of YOLOv4. YOLOv5 introduced a greater number of CSP structures in the neck than YOLOv4, resulting in a larger neck with more feature fusion capability and a reduction in the inference speed. DET-YOLO was faster than YOLOv5 because a large number of skip connection structures were discarded, and the use of FFCN sped up the calculation of DEViT. Future research should focus on how to enhance self-attentive modules and on embedding methods to increase speed.

As shown in Figure 10, we visualized the prediction results on DOTA, RSOD, and UCAS-AOD and compared them with YOLOv4 and YOLOv5 to demonstrate the predictive effect of DET-YOLO.

### 4.2. Ablation Experiments

In order to confirm the efficacy of the modifications we made, we conducted ablation experiments on each component; the results are presented in the Table 8. We used YOLOv4 as our baseline. After converting the backbone from cspdarknet53 to ViT, it was discovered that mAP decreased. This was due to the fact that ViT uses a crude linear embedding layer to convert images to high-dimensional embeddings, thereby destroying a substantial amount of effective feature information. After FPN was replaced with DSDP, the mAP increased by 0.014. This was a result of DSDP, which focuses the network on the target area and mitigates, to some extent, the target loss caused by cutting during the embedding process. However, after replacing linear embedding with deformable embedding, the mAP was greatly increased by 0.037. This was due to the fact that the cutting window could be deformed adaptively during the embedding procedure, effectively preserving the feature data. In comparison to ViT, which employs deformable embedding rather than linear embedding, DEViT’s spatial feature extraction capability was enhanced by its use of FCFN rather than FFN. Consequently, the mAP increased by 0.023. Using DSDP instead of FPN based on DEViT increased the mAP by 0.018, the same as before. We thoroughly demonstrated the effectiveness of the proposed modules. Among them, deformable embedding improved the mAP by approximately 0.04; followed by FCFN, which improved the model accuracy by approximately 0.02; and DSDP, which improved the model accuracy by approximately 0.02. In addition, we showed the effect of replacing FCFN with a self-attention transformation (SAT) and DSDP with a pyramid attention layer (PAL) in DET-YOLO, as shown in rows 6 and 7. The mAP using SAT was 0.717, which was a decrease of 0.011 compared to using FCFN. The mAP using PAL was 0.720, which was a decrease of 0.008 compared to DSDP.

## 5. Conclusions

On the basis of YOLOv4, we developed DET-YOLO to address the issues of diverse target sizes, complex background, and the large number of small targets in aerial images. We improved YOLOv4 from two perspectives, the network for feature extraction and the network for feature fusion, in order to better adapt it to the characteristics of aerial images. On one hand, we proposed DEViT to replace CSPDarknet53 as the network for feature extraction in order to improve the global capability for feature extraction. DEViT uses deformable embeddings instead of linear embeddings to effectively reduce the loss of feature information when processing targets at different scales compared to conventional vision transformers. We used FCFN rather than FFN in the DEViT encoder block to improve the extraction of location information while simultaneously reducing computational effort. On the other hand, we made a replacement for the FPN in the neck. As an alternative to the FPN, we proposed DSDC for extracting multiscale features from a single-scale feature map. DSDC is capable of adaptively focusing on key regions in order to enhance the ability to extract information about desired feature targets. DET-YOLO was evaluated using the widely used DOTAv1.5, RSOD, and UCAS-AOD datasets. For these datasets, the mAP values of the method proposed in this paper were 0.728, 0.952, and 0.945, which were superior to YOLOv4. The preceding results conclusively demonstrated the efficacy of the proposed DET-YOLO algorithm for object detection in aerial images. Experiments with ablation demonstrated that each of the proposed modules had a practical enhancement effect. In general, our proposed DET-YOLO model is a detector suited for aerial target detection tasks. In a subsequent work, we will enhance the linear embedding process of the transformer and attempt to apply deformable embedding to the hierarchical transformer model, thereby making the transformer model more suited to specific tasks in aerial imagery.

## Figures and Tables

**Figure 1 sensors-23-02522-f001:**
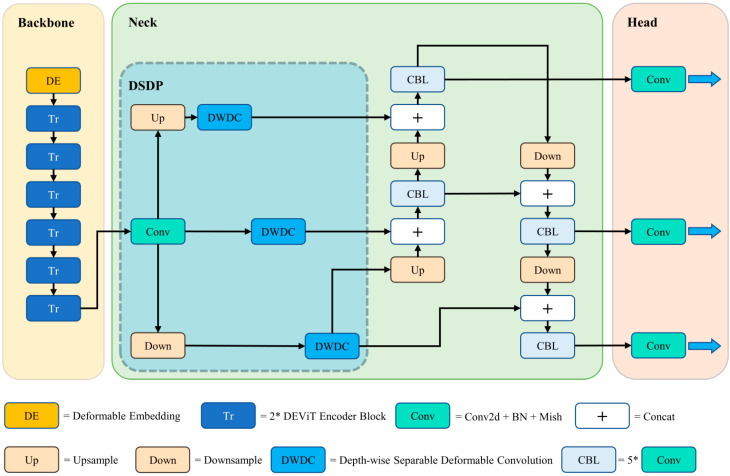
An overview of DET-YOLO. We use DEViT as the backbone, as opposed to CSPDarknet53 in the original implementation. In addition, the DSDP module is utilized in place of skip connections and SPP in order to obtain multiscale feature maps.

**Figure 2 sensors-23-02522-f002:**
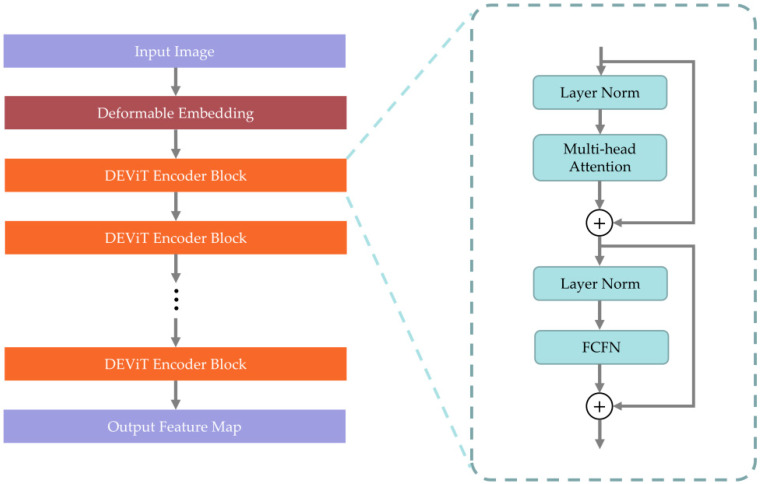
The DEViT architecture.

**Figure 3 sensors-23-02522-f003:**
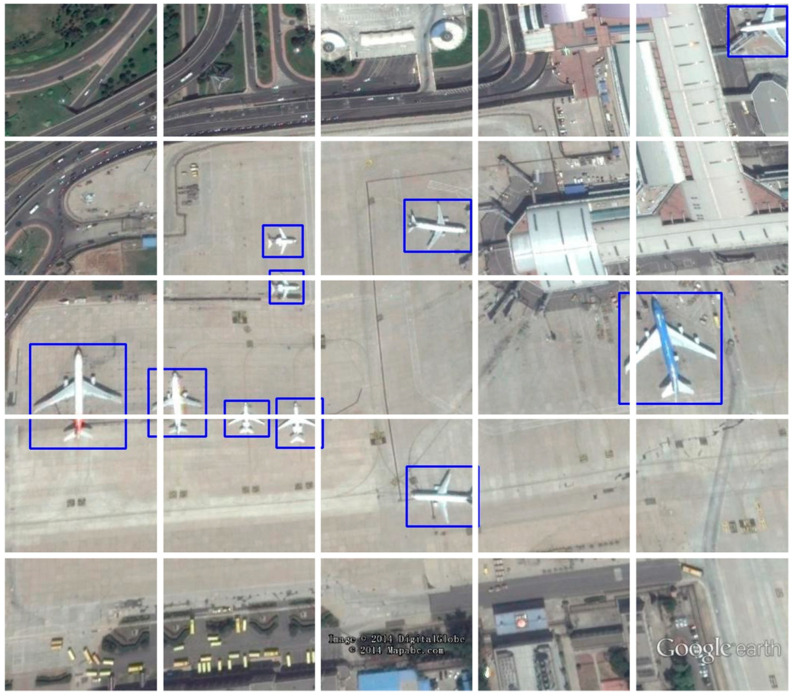
Vanilla patch embedding example. Patch splitting is represented by the white lines, while the ground-truth boxes are represented by the blue lines. Some targets’ semantics are compromised.

**Figure 4 sensors-23-02522-f004:**
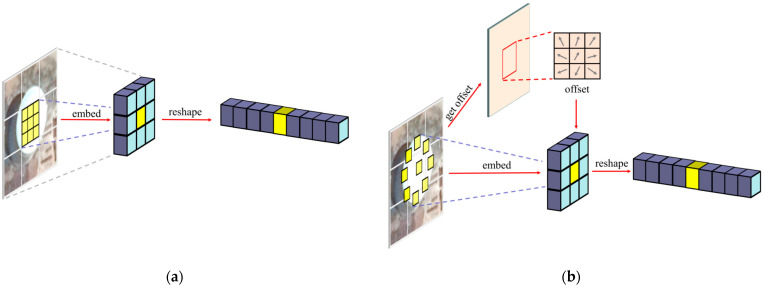
Comparison of vanilla patch embedding and deformable embedding: (**a**) vanilla patch embedding and (**b**) deformable embedding.

**Figure 5 sensors-23-02522-f005:**
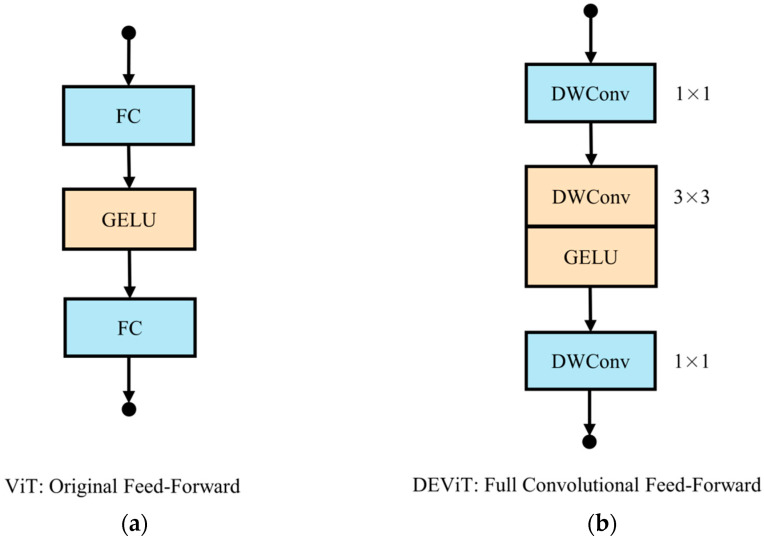
Comparison of FFN and FCFN: (**a**) the vanilla FFN of a vision transformer; (**b**) the FCFN of DEViT.

**Figure 6 sensors-23-02522-f006:**
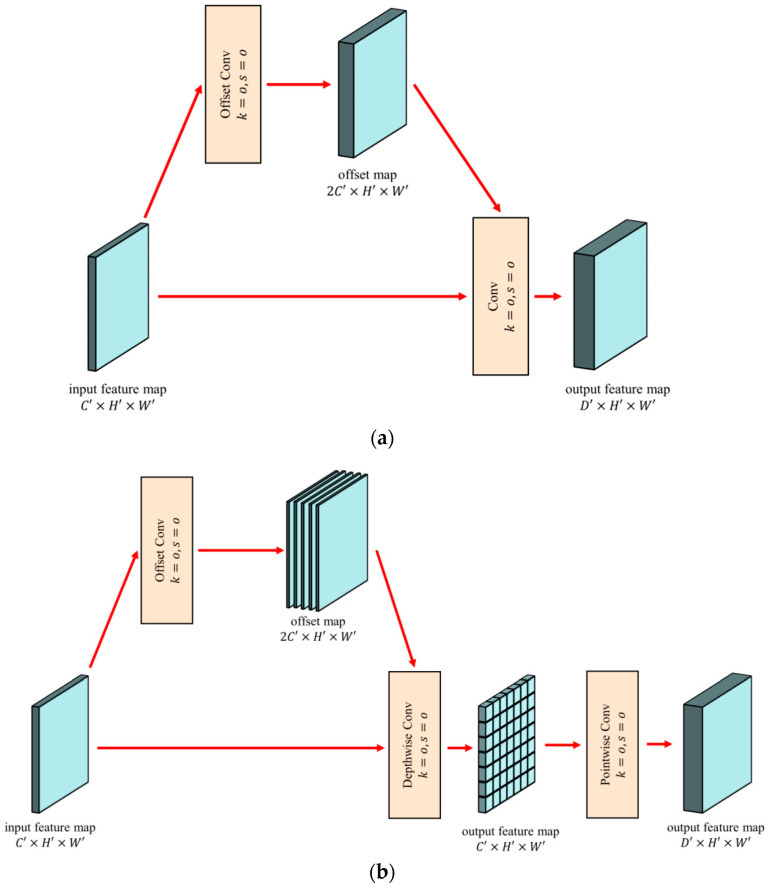
Comparison of vanilla deformable convolution and depthwise separable deformable convolution: (**a**) vanilla deformable convolution; (**b**) depthwise separable deformable convolution.

**Figure 7 sensors-23-02522-f007:**
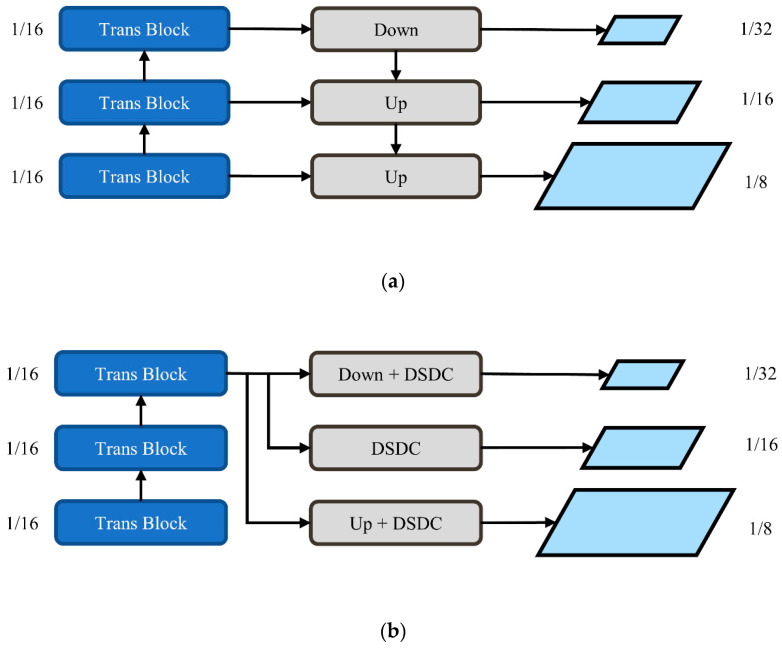
Comparison of feature pyramid network and depthwise separable deformable pyramid module: (**a**) FPN; (**b**) DSDC.

**Figure 8 sensors-23-02522-f008:**
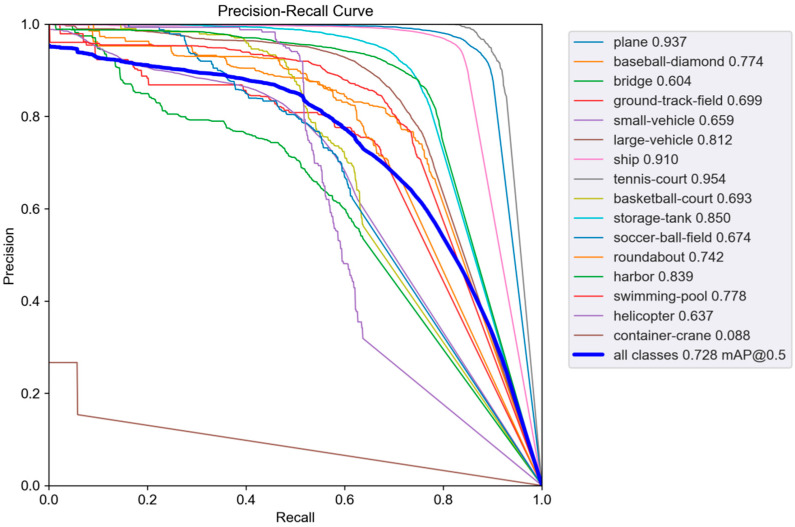
The P-R curve of DET-YOLO on the DOTA dataset at an IoU threshold of 0.45 and a confidence threshold of 0.25.

**Figure 9 sensors-23-02522-f009:**
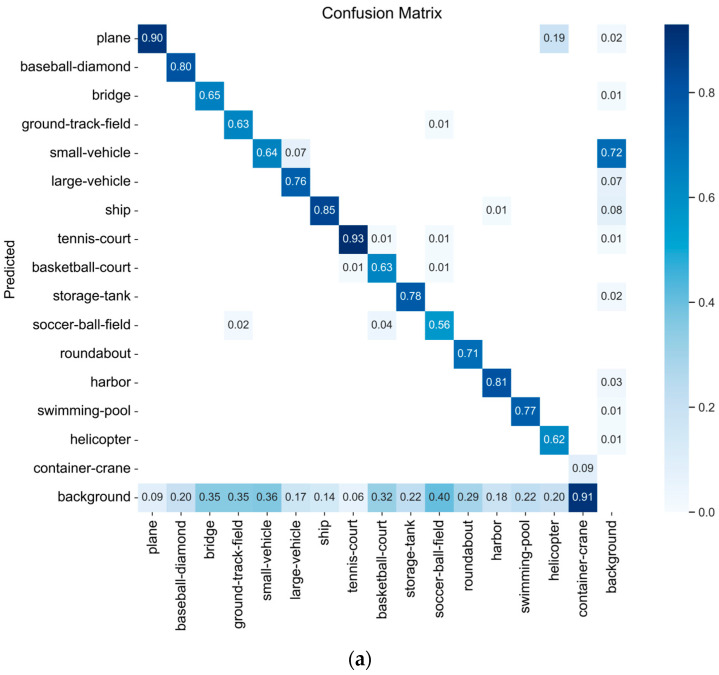
Confusion matrix of DET-YOLO on the DOTA dataset at an IoU threshold of 0.45 and a confidence threshold of 0.25: (**a**) Confusion matrix based on the class of the target; (**b**) Confusion matrix based on the size of the target.

**Figure 10 sensors-23-02522-f010:**
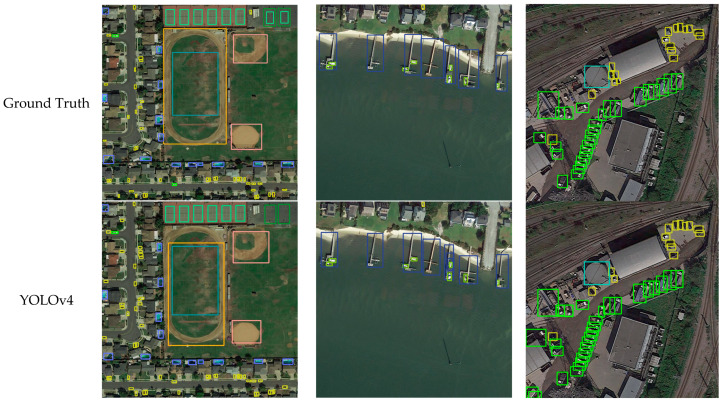
Experimental results for (**a**) DOTA dataset; (**b**) RSOD dataset; (**c**) UCAS-AOD dataset; and (**d**) UAVDT dataset.

**Table 1 sensors-23-02522-t001:** Comparison of instance size distributions of the RSOD, UCAS-AOD, and DOTA datasets.

Datasets	Below 10 Pixels	10–50 Pixels	50–300 Pixels	Above 300 Pixels
RSOD	0%	36%	60%	4%
UCAS-AOD	0%	32%	68%	0%
DOTA	16%	69%	14%	1%
UAVDT	1%	89%	10%	0%

**Table 2 sensors-23-02522-t002:** Division of the RSOD, UCAS-AOD, and DOTA datasets.

Datasets		Class	Image Number
RSOD	Training Set	Oil Tank	102
		Aircraft	270
		Overpass	140
		Playground	140
	Test Set	Oil Tank	63
		Aircraft	176
		Overpass	36
		Playground	49
UCAS-AOD	Training Set	Aircraft	600
		Car	310
	Test Set	Aircraft	400
		Car	200
DOTA	Training Set	-	14,729
	Test Set	-	5066
UAVDT	Training Set	-	23,829
	Test Set	-	16,580

**Table 3 sensors-23-02522-t003:** Comparison of the performances on the DOTA dataset.

Methods	*p*	R	mAP (IOU = 0.5)
Faster R-CNN	0.710	0.594	0.631
SSD	0.696	0.522	0.587
RetinaNet	0.714	0.585	0.622
YOLOv3	0.716	0.532	0.575
YOLOv4	0.732	0.593	0.653
YOLOv5	0.742	0.607	0.659
TPH-YOLOv5 [49]	0.785	0.643	0.683
SPH-YOLOv5 [47]	**0.806 ***	**0.683**	0.716
DET-YOLO	0.748	0.668	**0.728**

* Bold indicates the best result in the current table. Table 4, Table 5 and Table 6 are the same.

**Table 4 sensors-23-02522-t004:** Comparison of the performances of various models on the RSOD dataset.

Methods	*p*	R	AP				
			Aircraft	Oil Tank	Overpass	Playground	mAP (IOU = 0.5)
Faster R-CNN	0.873	0.748	0.859	0.867	0.882	0.904	0.878
SSD	0.824	0.682	0.692	0.712	0.702	0.813	0.729
RetinaNet	0.893	0.846	0.867	0.882	0.817	0.902	0.867
YOLOv3	0.850	0.693	0.743	0.739	0.751	0.852	0.771
YOLOv4	0.903	0.735	0.855	0.858	0.862	0.914	0.872
YOLOv5	0.897	0.872	0.873	0.884	0.854	0.932	0.886
DET-YOLO	**0.925**	**0.909**	**0.925**	**0.963**	**0.918**	**1.000**	**0.952**

**Table 5 sensors-23-02522-t005:** Comparison of the performances of various models on the UCAS-AOD dataset.

Methods	*p*	R	AP		
			Airplane	Car	mAP (IOU = 0.5)
Faster R-CNN	0.896	0.775	0.873	0.865	0.869
SSD	0.770	0.574	0.702	0.726	0.714
RetinaNet	0.887	0.742	0.843	0.865	0.854
YOLOv3	0.772	0.692	0.757	0.756	0.757
YOLOv4	0.894	0.732	0.857	0.862	0.859
YOLOv5	0.892	0.785	0.892	0.858	0.875
DET-YOLO	**0.962**	**0.863**	**0.997**	**0.892**	**0.945**

**Table 6 sensors-23-02522-t006:** Comparison of the performances of various models on the UAVDT dataset.

Methods	*p*	R	AP			
			Car	Truck	Bus	mAP (IOU = 0.5)
YOLOv4	0.438	0.431	0.765	0.104	0.332	0.400
YOLOv5	**0.471**	0.427	0.767	0.121	0.349	0.419
DET-YOLO	0.464	**0.432**	**0.777**	**0.131**	**0.365**	**0.424**

**Table 7 sensors-23-02522-t007:** A comparison of the inference times for various methods on DOTA datasets.

Methods	Speed (ms per Picture)
YOLOv3	28.4 ms
YOLOv4	43.2 ms
YOLOv5	102.0 ms
TPH-YOLOv5	123.5 ms
DET-YOLO	62.1 ms

**Table 8 sensors-23-02522-t008:** The results of ablation experiments on DOTA datasets.

Methods	ViT	DE	DEViT	FPN	DSDP	SAT	PAL	mAP_0.50_	mAP_0.50:0.95_
YOLOv4	√							0.653	0.438
DET-YOLO	√			√				0.650	0.433
	√				√			0.664	0.447
	√	√		√				0.687	0.471
			√	√				0.710	0.495
	√	√			√	√		0.717	0.509
			√				√	0.720	0.514
			√		√			0.728	0.515

## Data Availability

Not applicable.

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
