# Peer review of "YOLOv4 with Deformable-Embedding-Transformer Feature Extractor for Exact Object Detection in Aerial Imagery"

_sensors, 2023, doi:10.3390/s23052522_

Round 1
Reviewer 1 Report
Authors propose incremental improvements on YOLOv4. Results are promising, but novelty is limited, and experimentation due to a lacking of realistic testing on more datasets.> we propose deformable embedding instead of linear embedding, and propose full convolution feedforward network (FCFN)
No comparison on why FCFN would be better than self-attention transformation. In high level feature space it's more likely attention would perform better than convolutions.
> depth direction separable deformable pyramid module (DSDP)
No comparison with Pyramid attention layer.
> Our experiments on DOTA, RSOD and UCAS-AOD data sets show that the average accuracy 18 (mAP) of our method reaches 0.728, 0.952 and 0.945 respectively
No benchmark on UAVDT/SeaDronesSee datasets which are main benchmark for object detection from aerial images.
No open-source and reproducible code.
pram -> param
> 50/150 epochs.
Why such selection, how was selected stopping condition?
Author Response
Thank you very much for your review report. All of your suggestions and comments for the manuscript are very important, they have an important guiding significance for my paper writing and research work.

Reviewer 2 Report
The paper is well-written. The literature build-in introduction needs little improvement. Train and test optimization is justified well.
Author Response
Thank you very much for your review report. Thank you for your understanding and recognition of our work. We noticed that you selected the "Moderate English changes required" option. Our article has be handed over to the language editing services for English revisions. Thank you for your suggestion!
Reviewer 3 Report
Dear Authors:
Your paper: “YOLOv4 with Deformable Embedding Transformer Feature Extractor for Exact Object Detection in Aerial Imagery” presents a recurrent and interesting theme: extracting information from ultra-high spatial resolution images. Although you positioned your work as a solution for detecting targets in aerial images, you used datasets that have orbital images with high spatial resolution. However, aerial images usually have higher spatial resolutions than those available from orbital platforms. In this context, it is interesting to reconsider the use of the term aerial images. In my understanding, the term digital images with ultra-high spatial resolution would be more coherent and would adequately cover the study carried out. You could take advantage of this concept and also mention the images obtained by UAV that have centimeter resolutions.
Regarding the writing of the article, please implement a review using the help of a native English speaker. I pointed out 18 comments to that effect in the digital file. I also request your attention to the following questions:
1) Lines 62 to 76 are very similar to the article objectives shown in lines 81 to 98.
2) Figure 4: Compress the distribution of parts a and b in the Figure.
3) Lines 290 to 308: The study proposes a methodology applicable to aerial images. However, the datasets employed are derived from orbital platforms. It would be interesting to better contextualize this issue because in aerial images the expected spatial resolution is a few centimeters. In the datasets, the scenes have a spatial resolution in decimeters.
4) Lines 309 to 312: Are these object size classes based on another study or did you define them? Please mention the criteria for this classification.
5) Lines 316 to 321: detail the criterion for defining the sample size for training and validation.
6) Lines 340 to 348: cite the bibliographical references used to define the parameters of the network.
7) Tables 2, 3, and 4: highlight with bold font the best results for each column.
8) Figure 9: What if the confusion matrix analysis were performed according to the size of targets? Very small, small...
9) Line 401: Discuss this situation in more depth.
10) Figure 10: if instead of presenting the label with the description of the target, you created a caption and removed the label from the figure? This would make the comparison easier.
11) Review the conclusions and seek to respond to the study objectives presented in lines 80 to 90.
I ended my review by congratulating them for the work and article presented.
Respectfully,

Author Response

(The authors gave the same response as above.)

Round 2
Reviewer 1 Report
Manuscript improved significantly.
All notes were taken into account.
Author Response
Dear Reviewer
Thank you very much for your review report. Thank you for your understanding and recognition of our work. Thank you for your good suggestions and comments again.
Sincerely yours,
YiHeng Wu
Jianjun Li
Reviewer 3 Report
Dear Authors
The new version of your manuscript: "YOLOv4 with Deformable Embedding Transformer Feature Extractor for Exact Object Detection in Orbital Imagery" presented several changes that, in my understanding, made reading the article more fluid and assimilable. If I may, I would add the term ultra-high spatial resolution orbital imagery to the title of the article or in the keywords.
The alterations produced in the images made the interpretation of the results easier.
Comparing with the first version, the article now has merit to be published.
I conclude by congratulating them for the work and for the article presented.
Respectfully,
Author Response
Dear Reviewer
Thank you very much for your review report. Thank you for your understanding and recognition of our work.
Comment : If I may, I would add the term ultra-high spatial resolution orbital imagery to the title of the article or in the keywords.
Response : Thank you for your suggestion! We have added the term ultra-high spatial resolution orbital imagery to the keywords.
Thank you for your good suggestions and comments again.
Sincerely yours,
YiHeng Wu
Jianjun Li